## Research Article

stressful event; depression; anxiety; older adult

**Corresponding author:**
Oscar Flores-Flores MD, PhD;
Emails: ofloresf@usmp.pe

# *"How I would like to forget"*: Lived experiences of traumatic and stressful life events among older adults in Peru

Alejandro Zevallos-Morales[1,2], Gabriela Ramos-Bonilla[3], Lorena Rey[1,2], Ivonne Carrión[1], Diego Otero-Oyague[2,4], Trishul Siddharthan[5], John R. Hurst[6], José F. Parodi[1], Joseph J. Gallo[7], Suzanne L. Pollard[8] and Oscar Flores-Flores[1,2] [iD]

[1]Facultad de Medicina Humana, Centro de Investigación del Envejecimiento (CIEN), Universidad de San Martín de Porres, Lima, Perú; [2]Asociación Benéfica PRISMA, Lima, Perú; [3]Facultad de Ciencias Sociales, Grupo de Investigación Edades de la Vida y la Educación (EVE), Pontificia Universidad Católica del Perú, Lima, Perú; [4]Facultad de Psicología, Grupo de Investigación en Psicología Comunitaria (GIPC), Pontificia Universidad Católica del Perú, Lima, Perú; [5]Division of Pulmonary and Critical Care, Miller School of Medicine, University of Miami, Miami, FL, USA; [6]UCL Respiratory, University College London, London, UK; [7]Department of Mental Health, Johns Hopkins University Bloomberg School of Public Health, Baltimore, MD, USA and [8]Center for Global Non-Communicable Diseases, School of Medicine, Johns Hopkins University, Baltimore, MD, USA

## Abstract

Traumatic and stressful life events can have lasting effects on mental health, particularly among older adults in low-resource settings. In Latin America, there is limited qualitative evidence capturing the lived experiences of these events. This study explores how older adults in Peru reflect on traumatic and stressful events throughout their lives, and how these experiences continue to shape their mental health in later life. This qualitative study was nested within the Global Excellence in COPD Outcomes (GECo) study in Lima, Peru. We conducted semi-structured, narrative-based interviews with 38 older adults (≥60 years) with moderate to severe symptoms of depression (Patient Health Questionnaire-9 ≥ 10), anxiety (Beck Anxiety Inventory ≥ 16) or a history of mental health treatment. Four main categories emerged: (1) violence (emotional, physical or sexual), (2) abandonment or loss of close relatives, (3) onset of severe illness or disability and (4) other miscellaneous life disruptions. Participants described their memories of past stressful events as deeply embedded in current thoughts and, in some cases, as shaping how they experience certain emotions in the present. Addressing trauma in older adults may improve well-being in low-resource settings. Recognizing the enduring impact of life-course stressors is crucial for culturally sensitive mental health interventions.

## Impact statement

This study explored the traumatic and stressful events experienced across the life course of Peruvian older adults and their influence in current symptoms of depression and anxiety. This study highlights the need for improvements in the identification of trauma across the different life stages. Furthermore, proper management of these experiences may improve outcomes when reaching older adulthood. Finally, this study serves as a valuable resource which can inform culturally adapted interventions in this age group.

## Introduction

Trauma episodes emerged spontaneously as an important issue during open-ended interviews in a study of depression and anxiety expression among Peruvian older adults from low-resource settings (Flores-Flores et al., 2020). Our purpose, during those interviews, was to lay the foundation for mental health interventions tailored to the needs of older adults from low-resource settings in Peru and similar contexts (Cruz-Riquelme et al., 2024). Approaches with a narrow focus or use of a biomedical model may overlook important social, contextual and cultural factors (Armelagos et al., 1992). Furthermore, interventions that implement cultural understanding of illness can lead to improved engagement and better health outcomes (Napier et al., 2014; Subandi et al., 2021).

Our previous analysis characterized the experience of depression and anxiety among older adults, its causes, and their own strategies for coping (Flores-Flores et al., 2020). After identifying spontaneous traumatic and stressful events during our interviews we considered important for community-based intervention development, so we took the opportunity to examine in detail how participants integrated stories of trauma and stressful events with how depression and

anxiety were experienced. Because participants described traumatic experiences across their life course, we chose to retain a life course perspective in relating the findings. This approach not only helps trace the connections between events over time but also situates them within a broader sociocultural context, highlighting their cumulative effects and contributing to a deeper understanding of their complexity (Band-Winterstein and Eisikovits, 2009).

The link between trauma and conditions such as depression, dementia and abuse in later life is well-established (Frías, 2016; Brownell, 2019; Bruno and Saucedo, 2020; Ramos, 2020; Tani et al., 2020). However, in Latin America and the Caribbean (LAC), most studies take an epidemiological approach, with few qualitative studies capturing the lived experiences of those affected. In Peru, most existing research is epidemiological in nature. For example, studies in pregnant women have associated childhood abuse with later post-traumatic stress disorder and suicidal ideation (Zhong et al., 2016; Sanchez et al., 2017). Furthermore, in incarcerated population, the presence of adverse events like child abuse and caretaker drug consumption was linked to both depression and anxiety later in life (Castañeda Montenegro and Ascarruz Asencios, 2019). Within the wider LAC context, qualitative evidence also appears limited. One study in Colombia explored stressful life events among older adults, identifying living conditions, gender-based violence and lack of mental health awareness, as major topics shaping their mental health needs (Giebel et al., 2025).

Humans are natural storytellers, and personal narratives offer more than a record of events, they reveal how individuals make sense of their experiences, selecting key moments to craft coherent, meaningful stories (Kaufman et al., 2005; Young and Rodriguez, 2006; Tetley et al., 2009; Lindenmeyer et al., 2011). Thus, this study explores the perspectives of Peruvian older adults on traumatic and stressful life events, focusing on how they reflect on and interpret these experiences, rather than seeking an objective account of "what happened" to uncover crucial insights into decision-making and intervention opportunities over time.

## Methods

### Study sample

#### The parent study: global excellence for COPD outcomes

The qualitative study was nested within the Global Excellence in COPD Outcomes (GECo) study – a cross-sectional, population-based study that randomly recruited 3,500 participants aged 40 and older from urban districts in South Lima, Peru (Siddharthan et al., 2022). Participants primarily consisted of Andean migrants from low-resource settings who moved to Lima in the 1970s seeking better economic opportunities. The GECo study, conducted between 2018 and 2020, included the Patient Health Questionnaire-9 (PHQ-9) (Calderón et al., 2012) and Beck Anxiety Inventory (BAI) (Basto-Abreu et al., 2022), to assess depressive and anxiety symptoms, respectively. Individuals were excluded if they had active pulmonary tuberculosis, severe cognitive or physical impairments preventing spirometry, a recent myocardial infarction, or eye, thoracic or abdominal surgery within the past 3 months (Flores-Flores et al., 2020; Siddharthan et al., 2022).

#### The qualitative study

For the qualitative study (Flores-Flores et al., 2020), we purposively selected a sample of 38 older adults (60 years and older). Participants were chosen based on the presence of one of the three following criteria: (1) having moderate to severe symptoms of depression (PHQ-9 score of ≥10) (Calderón et al., 2012), (2) having moderate-to-severe anxiety symptoms (BAI score of ≥16) (Basto-Abreu et al., 2022) and (3) history of receiving mental health treatment, regardless of their current symptom status. During in-depth interviews, when asked about the causes or reasons for their depressive symptoms, participants described various traumatic and stressful life events. These events were not necessarily recent but occurred across different stages of their lives. For this paper, we focused on these events, the narratives surrounding them and the meanings older adults attributed to their experiences.

### Data collection

In-depth interviews with older adults were conducted between October 15, 2018, and February 1, 2019, by trained interviewers at the participant's home. Forty-four individuals were approached to participate in this study, five refused to participate due to lack of time or interest and one individual was excluded during the interview due to untreated schizophrenia. Our final sample was of 38 participants. Three of the interviewers had a medical background and one had a background in social sciences. All interviewers had prior training in qualitative methods, specifically in conducting in-depth interviews with older adults. Furthermore, interviewers were instructed to stop recording if at any point the participant felt overwhelmed, giving them time to calm and the option to finish the interview early. Finally, there was not direct supervision during the interviews; however, interviews were performed in pairs with a main interviewer and a second person usually helping with note taking. We applied semi-structured interviews with narrative approaches (Creswell et al., 2007; Victor et al., 2007; Spencer et al., 2013). The interview guide (Supplementary Material 1) was developed by the authors with the intention to explore illness experiences of depression and anxiety. The interviews lasted between 40 and 70 minutes. Very few older adults declined to participate in the interviews, and when they did, it was primarily due to scheduling conflicts or personal availability. Finally, sociodemographic characteristics (age, sex, level of education, marital status and employment status) were collected during the parent GECo study.

### Analysis of data

This secondary analysis builds upon the primary analysis, which explored older adults' perceptions toward depression and anxiety (Flores-Flores et al., 2020). Audio recordings and transcriptions were securely shared between a professional transcriber and authors via encrypted files. Our multidisciplinary team of researchers (OF-F, AZ-M, LR, IC, SP, JG and GR) conducted the analysis of the interviews using an inductive thematic analysis strategy. Coding was conducted inductively and iteratively, and all codes and emerging themes were discussed among team members to ensure conceptual clarity and consistency. Discrepancies were resolved through group discussion until consensus was reached. Subsequently, a subgroup of researchers (OF-F, AZ-M, GR, LR and DOO) focused on identification of adverse, traumatic and violent events narrated by the participants across the life course. We focused on events that older adults mentioned as possible reasons or explanations for their symptoms of depression and anxiety. This information emerged from the analysis of the full interview, but was primarily drawn from the following questions: 3.1: How do you feel when you are depressed? 3.2: When did those feelings start? and 3.3: Why do you think these feelings started? These questions provided most of the information related to

stressful or traumatic events. We grouped these events into four main categories: violent events (emotional, physical or sexual), abandonment or loss of close relatives, onset of severe illness or disability and other miscellaneous events. Finally, we sought to identify connections between those adverse situations and discuss the meaning of those situations and descriptions of the current mental health status. To facilitate data organization and analysis, MAXQDA software (VERBI GmbH, Berlin Germany, Version 20.2.2) was used (Clarke and Braun, 2013).

## Ethical considerations

Ethical approval was obtained from the Institutional Review Boards at Johns Hopkins University and A.B. PRISMA. Researchers in the study completed a training course in ethics and human subject protections. certified by the CITI program. Oral informed consent was obtained from participants. Fictitious names are used in the selected quotes to protect participant's identities.

## Results

### Sample characteristics

We analyzed 38 interviews and found narratives related to stressful events in 28 of them. Participant characteristics are described in Table 1. Among these 28 participants, 19 (68%) were women and 9 (32%) were men. The average age of the participants was 67.1 ± 6.7 years. Fourteen participants were married, and all lived with relatives in the same household. Additionally, 24 (86%) participants exhibited symptoms of depression or anxiety (PHQ-9 score ≥ 10, BAI score ≥ 16), and five participants were receiving mental health treatment.

### Traumatic and stressful events over the life course

In this section, we present the traumatic and stressful events across different life stages – youth, middle age or old age – examining their characteristics, context and meanings from the perspectives of older adults. Figure 1 summarizes the types of events found through the described life stages.

### Youth

Most participants identified events that happened during their childhood or adolescence. Even decades later, these events were perceived as sources of sadness and anguish. Commonly mentioned situations included physical, emotional, occupational and sexual abuse, primarily perpetrated by family members and, to a lesser extent, by other individuals to whom they were not related:

> *"Yes [I was] eleven, twelve years old. From that age, I was separated from my mother, I came here to Lima (capital city in Peru) [to work as a domestic worker], and it did not go well for me there, because the lady (mother of the employer) began to hit me (cries)."* Vanessa, 62 years old

Some participants, both male and female, indicated that were forced to work during their early years. In some cases, this work was described as an act of exploitation (labor violence) and was often accompanied by other types of physical and emotional violence. Those situations sometimes made impossible to access opportunities for study and personal development, something they still regretted.

**Table 1.** Social demographics of 38 participants in Lima, Peru, 2018–2019

| Characteristics | Overall (N = 38) |
|---|---|
| Age in years (mean ± SD) | 68.0 ± 7.8 |
| Sex, n (%) | |
| Female | 23 (60.5) |
| Male | 15 (39.5) |
| PHQ–9 (mean ± SD) | 9.97 ± 5.84 |
| BAI (mean ± SD) | 16.84 ± 7.23 |
| Education level, n (%) | |
| Higher | 2 (5.2) |
| Secondary (complete or incomplete) | 20 (52.6) |
| Primary (complete or incomplete) | 14 (36.8) |
| None | 2 (5.3) |
| Comorbidities, n (%) | |
| Hypertension | 19 (50.0) |
| Diabetes | 7 (18.0) |
| Asthma | 6 (15.8) |
| Arthritis | 6 (15.8) |
| Employment status, n (%) | (n = 34) |
| Employed | 17 (50.0) |
| Unemployed | 17 (50.0) |
| Civil status, n (%) | (n = 32) |
| Married | 15 (46.9) |
| Separated, divorced or single | 8 (25.0) |
| Widower | 9 (28.1) |
| People at home, n (%) | (n = 37) |
| Alone | 1 (2.7) |
| Partner and other people | 13 (35.2) |
| Without partner and other people | 23 (62.1) |
| Medication for depression and/or anxiety, n (%) | (n = 32) |
| Yes | 7 (21.9) |
| No | 25 (78.1) |
| State of depression and/or anxiety, n (%) | |
| Only depressive symptoms (PHQ–9 ≥ 10) | 11 (28.9) |
| Only anxiety symptoms (BAI ≥ 16) | 11 (28.9) |
| Both depressive and anxiety symptoms | 11 (28.9) |
| No depressive or anxiety symptoms | 5 (13.3) |

*Note:* Quantitative data was obtained from the parent study, Global Excellence in COPD Outcomes (GECo). Due to missing values in some variables, the totals presented may not correspond to the full sample size. Percentages calculated among participants with available data. BAI, Beck Anxiety Inventory; PHQ-9, Patient Health Questionnaire-9; SD, standard deviation.

> *"I came alone [to Lima], to look for a future. I said: 'to study, not to work'. But I didn't get to study much because the job I had [didn't give me time]. And so, I only [completed] the third grade of primary school. […] to this day, I continue to work, to be able to help them (their parents), because back in the province I lived in the mountains, [there] there was not much help."* Benedicta, 60 years old

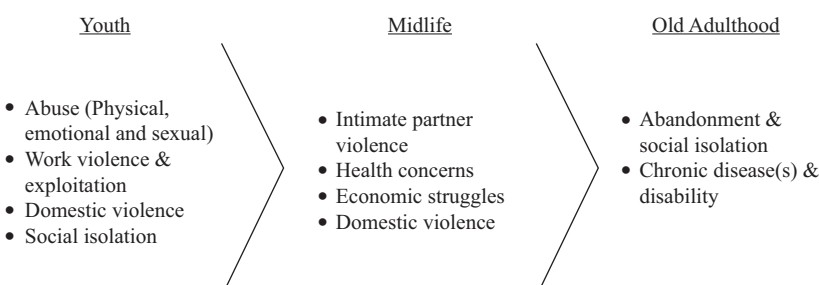

**Figure 1.** Diagram of stressful events over the life course emerged from interviews.

For many participants, working occurred in the context of family separation due to the death of a guardian or abandonment. Moving away from their families or places of origin at a very young age meant sadness, fear, and loneliness. This early displacement also left them vulnerable and unprotected, exposing them to situations of violence and abuse.

> *"I lived [in one Andean region] until I was nine years old, or so. From there, a cousin who lived in [Lima] brought me, and that year she brought me to take care of her little children. (…) She mistreated me a lot. My mom found out [because] I sent her [a] letter with a [person] who was going to travel with her and told her [mom] that she was mistreating me. [Later] she sent me a juvenile judge to have me taken to the police station, from the police station I returned to [land of origin]."* Judith, 66 years old

In some cases, participants chose to leave home to protect themselves from domestic violence. Vanessa, 62, who lost her father in childhood and was raised by the owner of the farm where her father had worked, shared her story of struggle and the need for her mother's love.

> *"Well, my childhood was very, how could you say?, not so safe, well my mommy, my dad passed away [when I was] very young, he left me very young…about three years old, more or less, but I do remember the day that my father died, he was poisoned and died in front of us, (…), my mother had baptized me with the landowner [godmother]., She took care of me, more or less until I was old enough to understand, I realized that she was not my mother, I looked for my mother. Obviously, I preferred my mother even though the landowner raised me, with [housewives], with, but I was looking, I felt the love that I needed my mother."* Vanessa, 62 years old

However, some participants did not perceive their work experiences as being entirely negative. Some participants valued their effort and ability to adapt to various jobs as key factors to get ahead.

> *"Yes, I have worked at home (as a housekeeper), I sold snowcones, then I worked in a bakery, I put myself to work in everything. [Also] I gave 'pension' (meals for workers). [Those are] thing[s] that one has to do to stand out, [one] has to fit into any job. I have worked all my life. Since I had the use of reason since I was a child."* Jimena, 62 years old

Regarding the experiences of sexual violence, it was observed that these cases were only identified and narrated by women. Women explained that those who perpetrated this type of abuse were mostly members of their families. Participants mentioned that their parents allowed or facilitated the abuse, which generated pain and led to the breakdown of family relationships. In the following quote, a participant who was abused by her stepfather and is currently experiencing depression explains how this situation affects her relationship with her children, who were conceived as a result of the assault.

> *Interviewer: "And when you were depressed, was there anyone who asked you how you felt?*
>
> *"No, no one. Not even my family, as I have already told you. (…)That is irrelevant but I practically… I was raped by my stepfather, (…) so I would hate my mom because my mom stood up for him. My stepfather was in prison. (…) perhaps [that is why] until now, as I say… the affection I show my children is very dry. I do not show them affection as other moms."* Noelia, 61 years old

## Midlife

During the adult life stage, the interviewees identified stressful events related to intimate partner violence (IPV), financial and/or work struggles, and the onset of severe health problems. Some women reported partner violence, mainly psychological, financial and physical. Usually, those types of violence were combined, as in the following case, when the participant's husband refused to give her money to access the oncological medical care she needed:

> *"You only have to trust in God because (…) I have trusted [erroneously, in other people], because I once asked my husband: 'Give me money for my ride, to get my tests (medical analysis).' [He told me:] 'Why am I going to give you [money] if you're going to die. Why don't those cancer patients get killed? Just kill yourself - that is what he told me. 'Who are you to tell me I'm going to die or live? In the name of Jesus, I am going to live', [I replied]. And here I am. I was 40 years old, 41 years they have detected that disease, and until now, I am living. I am already 73 years old. The Lord does not forget anything."* Esther, 72 years old

Most of the women interviewed shared that they had not reported the aggressions to the authorities, which contributed to their prolonged exposure to abusive situations throughout much of their adult lives. One participant recounted enduring repeated abuse by her husband over the course of their marriage and revealed that her fourth daughter was conceived as a result of rape.

> *"The time came when I did separate from the father of my children because I had the fourth [child] there, practically a fight and everything, because he was very 'macho', very possessive. I didn't just give in either, but you know, with a man's strength, there's only so much one can do, the thing is that he raped me and from that relationship my fourth daughter was born, which I assumed responsibility for."* Fiorella, 67 years old

Several participants of both sexes identified stressful events related to serious health problems experienced during adulthood but together with other events of IPV in a context of economic need:

> *"Just at that time I got sick in the lungs, TB [Tuberculosis]. I hadn't even realized how I was doing because I only lived to work for my children. Work, work for them. To work because my girls' father was*

*very irresponsible, he hit me and all that until [I went work in a company and I met a friend]. 'Hello' -he says- 'how are you? (…) Friend, I see you don't look well. Do not you realize that you are very thin? (…) I listened to him. The next day I went to the [health] post. They did checkups, everything. I think on the third day, the second day, they [health nurses] came from the [health] post, everything, they invaded my room. (…) And they detected that I had TB, and there, they forced me to live in an open field."* Vanessa, 62 years old

Economic problems were not only a cause of migration during childhood but also continued to be significant and frequently identified as stressful events by participants of both sexes. One participant recounts the stress and worry he experienced while trying to provide for his two families.

*"There are various stages of life [in] where depression has come. (…) First, because I had a double family: my previous children from my other partner and my children from here (current couple); and the responsibility was twofold. As a father, I have never wanted to abandon [my children]. That is, I have fully faced (…) my responsibility as a father on both sides, with my defects and virtues. But (…) that is where the worry begins, the depression. Sometimes [the money] is not enough."* John, 62 years old

### Older adulthood

One of the main stressful events identified by the participants in their current old adulthood was feeling abandoned by their families. This feeling of abandonment was not limited to physical separation but also included emotional distant, even when living in the same household.

*"You can take care of your mother, just as she had taken care of you when you were little. So, that [attention] is what is missing to our, older people, we get very depressed, because we feel alone, (continues crying), we have no one to care for us, no one that ask us if something hurt us?, if we feel something?, sometimes [they, (the children)] get bored."* Vanessa, 62 years old

On the other hand, a group of interviewees explained that their lives were being strongly affected by suffering from different chronic diseases that negatively affected functionality (e.g., visual loss), daily life, and social relationships. One of the most severe cases was a participant with end-stage renal disease who received dialysis every two days. In the following quote, he explained the burden that this has on his life.

*"I'm tired. I don't know. I ask other men [other patients], they have been on dialysis for fifteen, twenty years, how they have resisted. I have been four [years in dialysis], and I cannot take it anymore. I'm bored."* Frederick, 70 years old

Many participants shared how they had accumulated multiple traumatic experiences over their lives and how making sense of these events remains challenging. Several continue to experience distressing symptoms in the present. Carmen, 61, who suffered repeated sexual violence during childhood and endured a violent relationship with her partner for many years, expressed this ongoing struggle:

**"***How do you get out of that (depression)? Sometimes I wonder when one wants to forget, if you [could] forget. But I forget like this, [only] for the moment. I get distracted by one thing, another thing, and so on. But there are times when I am alone and [the memory] comes to mind, and then sadness, tears, even anger, rage come to me. So those things come to me. I ask, sometimes I say: how I would like to forget, how I would like to get these things out of my mind and heart, not to feel, right?… but I can't."* Carmen, 61 years old

## Discussion

While exploring the narratives surrounding the emergence of depression and anxiety symptoms among Peruvian older adults, significant traumatic stressful events were identified. Older adults vividly recounted past events that continued to exert a lasting impact on their mental well-being. Most of these distressing recollections occurred during their childhood and adolescence, with abuse and exploitation being the most reported traumatic experiences. In some cases, participants survivors of childhood abuse, went on to experience IPV during their midlife. Later in life, loneliness was a major issue among our participants; even in participants that reported living with family members.

Traumatic events during childhood or adolescence were the most frequent among our subjects. In LAC, around two in three children aged 1–14 experience domestic violent discipline (UNICEF, 2022). In Peru, according to the 2019 National Survey on Social Relations (ENARES), approximately 69% of children and 78% of adolescents had suffered some type of physical or psychological violence at home (Instituto Nacional de Estadistica e Informatica, 2020). In our sample, few older adults reported receiving help or protection at this point, or at any point during their life course. Early identification and adequate intervention could have stopped the cycle of violent events and likely reduced the occurrence of mental health problems later in life. Children that suffer trauma frequently experienced polyvictimization, continuing to endure various types of violence in subsequent stages of life (Frías, 2016; Wolfe, 2018).

Regarding events that happened during midlife, participants mainly discussed stressful events related to IPV, these included various types of violence: psychological, economic, physical and sexual. This burden of violence experienced by women throughout their lives can be attributed to a broader societal context characterized by deeply ingrained gender inequalities and associated roles (Costa et al., 2017). The same 2019 ENARES survey identified that 59% of Peruvians tolerate violence against women and 27% believe punishment is deserved after a woman disrespects her partner (Instituto Nacional de Estadistica e Informatica, 2020; Hirschfeld, 2025). Finally, prior experience of child abuse or sexual abuse has been linked to increased risk of IPV in pregnant women, emphasizing, the interconnectedness of these events during the life course (Barrios et al., 2015). In most cases, women continued with their abusive partner due to fear of losing financial support. IPV does continue into old age, and although physical abuse may decrease, psychological abuse persists along with the cumulative effect of decades of violence (Band-Winterstein and Eisikovits, 2009). Researchers suggest that to identify and understand IPV in later life, public health professionals and clinicians should focus on controlling behaviors and coercive control, which are commonly related to financial abuse (Roberto and McCann, 2021). While no male participant commented on events related to IPV, it is not possible to determine whether this is because men did not experience these situations or because they did not feel comfortable disclosing those events. However, these differences between men and women have been found in other studies (Kendler et al., 2001; Felitti, 2002; Axinn et al., 2013).

Toward more recent events, during their older adulthood, loneliness was the major stressful event reported. Currently, loneliness is a growing concern, especially among older adults, with severe consequences to their health (Berg-Weger and Morley, 2020). At old age, loneliness frequently took the form of physical, financial and emotional abandonment by their adult children. Furthermore,

loneliness was not as related to social isolation or family separation, as many individuals that lived with family members expressed loneliness. It should not be assumed that older people living with relatives cannot feel lonely (Ong et al., 2016). Previously, we stated that loneliness should be treated as an independent health problem throughout the life course (Flores-Flores et al., 2020) and not only as a symptom or precursor of depression.

Our study has several limitations. First, we may not have captured every individual's stressful event since the interview guide focused on exploring the illness experiences of depression and anxiety among Peruvian older adults, and not directly to stressful or traumatic events. Nevertheless, the events that emerged naturally from the interviews indicate how salient and meaningful these experiences are to the participants. Second, while the aim of a qualitative study is not generalizability, we acknowledge that our sample may not represent all older adults. It is possible that male participants were more reluctant to disclose certain experiences, particularly those involving sexual or emotional violence. However, we believe the interview setting fostered openness and trust, suggesting that while underreporting cannot be ruled out, the differences observed may also reflect real gendered patterns in the experience and narration of trauma. Qualitative samples are typically smaller than those in quantitative methods and often focus on outlying rather than normative cases. The most useful generalizations from qualitative studies are analytic, not "sample to population" (Miles, 2014). Qualitative sampling concerns itself with information power rather than representativeness (Malterud et al., 2016). Our sample consisted predominantly of women, mostly Andean migrants living in urban districts of Lima, so the experiences of older adults from other regions or those who did not migrate may differ. Third, although we ensured several aspects of rigor in qualitative research (Morse, 2015), including prolonged engagement by our research group, which has been working in this area for the last 5 years, we did not employ member checking by requesting participants' feedback on the findings.

Nonetheless, our study has several strengths. Interviews were conducted in the native language and in a private environment, ensuring comfort and confidentiality for the participants. As a research group, we frequently reflected on whether it was necessary to disclose the stories of violence mentioned in this paper. We concluded that sharing these stories was important to highlight events that had not been detected or reported by social or health services, which are detrimental to the well-being of Peruvian older adults. Furthermore, although it was not the primary aim of the study, many participants expressed gratitude for being listened to, emphasizing their need to be heard. After the interviews, we provided information about community resources for mental health care and referred participants to those services.

This study has important implications for public health interventions and research. First, unresolved early stressful events contribute to current feelings of sadness and distress in older adults. Therefore, it is crucial to detect and address these events throughout the life cycle to prevent recurrence and lifelong "scars." Abuse must be identified at various stages of life, and detection and care efforts should include not only the younger population but also older individuals who have the need and right to be heard. Although social and mental health services at the community level have increased in Peru compared to decades ago, few participants mentioned being aware of any social or health service agencies they could trust. Second, this study highlights the different nature of stressful events between men and women, which can help identify opportunities and tailor preventive interventions. Third, the

narratives of older adults provide insights that are essential for exploring experiences of violence in the Latin American region (Tsapalas et al., 2021). Finally, while we advocate for older adults to receive care to alleviate distressing thoughts, their experiential knowledge remains a valuable resource that can inform culturally adapted interventions (Cruz-Riquelme et al., 2024).

The older adults interviewed experienced a variety of stressful events throughout their lives, significantly impacting their health and life trajectories. Early life events remained vividly present in their thoughts and influenced their current feelings. While we did not specifically ask participants to recount their stressful events, these memories naturally emerged during discussions about their mood and feelings. This highlights the older adults' desire and need to be listened to. Social and health services must recognize and address the emotions tied to unresolved stressful events and consider the unique needs of older adults to improve their mental health and well-being.

**Open peer review.** To view the open peer review materials for this article, please visit http://doi.org/10.1017/gmh.2025.10060.

**Supplementary material.** The supplementary material for this article can be found at http://doi.org/10.1017/gmh.2025.10060.

**Data availability statement.** Qualitative data collected from this study is not available due to ethical considerations agreed with local IRB and participants in this study. The dataset used for this study can be obtained upon reasonable request to the corresponding author Oscar Flores-Flores (ofloresf@usmp.pe).

**Author contribution.** Alejandro Zevallos-Morales, Joseph Gallo, Trishul Siddharthan, Suzanne Pollard, John Hurst and Oscar Flores-Flores conceived and planned the study. Alejandro Zevallos-Morales, Oscar Flores-Flores, Ivonne Carrión and Lorena Rey conducted the interviews. Alejandro Zevallos-Morales, Gabriela Ramos, Lorena Rey, Jose Parodi, Diego Otero, Ivonne Carrión and Oscar Flores-Flores analyzed the results. Alejandro Zevallos-Morales, Gabriela Ramos and Oscar Flores-Flores wrote the manuscript. All authors contributed to the critical review of the manuscript. All authors approved the final version of the manuscript.

**Financial support.** Oscar Flores-Flores is supported by the National Institute of Mental Health (NIMH) and the Fogarty International Center (FIC) of the National Institutes of Health (NIH) under Award Number K43TW011586. The study was supported by the NIH Research Training Grant # D43 TW009340 from the Fogarty International Center, NINDS, NIMH and NHBLI. Suzanne Pollard was supported by a Mentored Research Scientist Development Award [1K01HL140048] from the National Heart, Lung, and Blood Institute, US National Institutes of Health. The content is the responsibility of the authors and does not necessarily represent the official views of the NIH. The GECo (Global Excellence in Chronic Obstructive Pulmonary Disease Outcomes) study was funded by the UK Medical Research Council (MR/P008984/1). Finally, the funders had no role in study design, data collection and analysis, decision to publish, or preparation of the manuscript.

**Competing interests.** None to declare.

**Ethics statement.** This qualitative study was approved by the Institutional Ethics Committee of the Asociación Benéfica Prisma (Peru; approval number CE0917.18) and the Institutional Review Board of School of Medicine from Johns Hopkins University (USA; IRB00007591). The ethical approval of the GECo study is documented elsewhere (Siddharthan et al., 2018; Siddharthan et al., 2022). Oral informed consent was obtained from participants, as well as permission for audio recording and the academic use of the information presented in this article. This approach was approved by both IRBs, given the minimal risk of the study, varying literacy levels among participants, and the preference to avoid signed documents in home settings. To address the sensitive nature of the topics discussed strategies were developed to support the emotional well-being of the interviewees. In some cases, interviews were paused to

allow participants to calm down and decide if they wished to continue. Additionally, fictitious names are used in the selected quotes to protect participant's identities.

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
