## [Reviewer Report · Review: 
*“How I would like to forget”*: Lived experiences of traumatic and stressful life events among older adults in Peru — R0/PR2]

This manuscript identified the traumatic experiences of older Peruvian adults who were experiencing symptoms of depression and anxiety with the goal of informing mental health interventions. The experiences were revealed during semi-structured interviews with a sample of participants from a larger study which was not specifically looking at trauma related experiences and late life impacts. This is an important and interesting contribution to the field and has the potential to inform meaningful public discourse, mental health intervention, and future research in this area.

This reviewer offers some suggestions to help strengthen the manuscript.

Abstract

The authors may wish to reframe the results (p. 3, lines 36-37) by clarifying that the participants identified the memories of past stress events as having negative effects on their current emotional state. As it reads now, it seems to imply causation which would not be possible with this study design.

Methods

Qualitative Study Section

Please identify which question(s) in the full interview were selected for analysis for this manuscript.

Analysis of data

Please describe the coding and thematic analysis process in more detail. How were themes identified? Was an inductive or deductive approach utilized? How were discrepancies among coders resolved? What was the percent agreement between coders? Which co-authors conducted the thematic analysis?

Did any participant refer to resilience, post-traumatic growth, or meaning making as a result of the difficult experiences? These themes would also be valuable to include and could serve to inform mental health interventions.

---

## [Reviewer Report · Review: 
*“How I would like to forget”*: Lived experiences of traumatic and stressful life events among older adults in Peru — R0/PR3]

Thank you for the opportunity to review this interesting paper on the lived experiences of older adults in Peru who have suffered traumatic experience. I have some comments and suggestions, which I hope will be helpful in improving the readability and quality of the paper.

Introduction

1) Line 64-68: It would be helpful if the authors could summarize the findings of the studies they cite, both from international contexts and from LAC.

Methods

2) Line 97-98: Did participants have to fulfil both the depression and anxiety cut-off criteria, or was fulfilment of one sufficient? Please explain this more clearly.

3) Line 108-109: What background did the interviewers have? How were they trained to handle difficult situations regarding the disclosure of traumatic experiences? Were they supervised? Please include this information.

4) Data collection (no concrete line): Please make it clearer that this is a secondary analysis of the data. It would also be helpful to know how many participants were approached and how many declined to participate in the interviews (and if so, for which reasons).

5) Please include the name of the ethical committee that approved the study and the approval numbers in the methods section. Please also state there how you obtained informed consent. Please also explain why only oral informed consent was obtained (was it audiotaped?)

Results

6) 136 ff: I was confused that the authors write that 38 participants took part in the study, but they only report the sociodemographic data for 28 persons who revealed traumatic experiences. I would suggest to either show the data for all 38 participants or state that the study includes 28 persons.

7) For data protection reasons I would suggest not to include the full first name to the quotes, but only an abbreviation

8) 238-139: I found the sentence in the quote difficult to understand. Maybe it can be slightly edited?

Discussion

9) As the authors have pointed out, the results of the study cannot draw conclusions about differences in traumatic events in women and men, because of the small sample size, and the fact that participants were not actively asked about traumatic experiences. It possible that men simply did not disclose about traumatic experiences/ sexualized violence.